# Evaluation of a Physical Activity and Multi-Micronutrient Intervention on Cognitive and Academic Performance in South African Primary Schoolchildren

**DOI:** 10.3390/nu14132609

**Published:** 2022-06-23

**Authors:** Johanna Beckmann, Siphesihle Nqweniso, Sebastian Ludyga, Rosa du Randt, Annelie Gresse, Kurt Z. Long, Madeleine Nienaber, Harald Seelig, Uwe Pühse, Peter Steinmann, Jürg Utzinger, Cheryl Walter, Markus Gerber, Christin Lang

**Affiliations:** 1Department of Sport, Exercise and Health, University of Basel, 4052 Basel, Switzerland; sebastian.ludyga@unibas.ch (S.L.); harald.seelig@unibas.ch (H.S.); uwe.puehse@unibas.ch (U.P.); markus.gerber@unibas.ch (M.G.); christin.lang@unibas.ch (C.L.); 2Department of Human Movement Science, Nelson Mandela University, Gqeberha 6011, South Africa; felicitas.nqweniso@mandela.ac.za (S.N.); rosa.durandt@mandela.ac.za (R.d.R.); madeleine.nienaber@mandela.ac.za (M.N.); cheryl.walter@mandela.ac.za (C.W.); 3Department of Human Nutrition and Dietetics, Nelson Mandela University, Gqeberha 6031, South Africa; annelie.gresse@mandela.ac.za; 4Department of Epidemiology and Public Health, Swiss Tropical and Public Health Institute, 4123 Allschwil, Switzerland; kurt.long@swisstph.ch (K.Z.L.); peter.steinmann@swisstph.ch (P.S.); juerg.utzinger@unibas.ch (J.U.)

**Keywords:** executive function, inhibitory control, information processing, randomized control trial, stunting

## Abstract

Executive functions (EFs) are essential for optimal academic development. Appropriate nutrition and physical activity (PA) have been shown to facilitate optimal cognitive development. Therefore, this study examined whether a 12-week school-based PA and multi-micronutrient supplementation (MMNS) intervention would improve cognitive and academic performance. A cluster-randomized controlled trial (RCT) was conducted. Children from four schools located in a peri-urban area of South Africa were randomly assigned to (i) PA + MMNS, (ii) PA + placebo, (iii) MMNS or (iv) placebo. Information processing and inhibitory control were measured with a computerized Flanker task. End-of-year results provided insight into academic achievement. Anthropometric measures were used to determine nutritional status. Data were analyzed with linear mixed-models, adjusting for baseline scores, school classes and age; 932 children (458 girls (49.1%), *M_age_* (mean age) = 8.42 ± 1.94 years) completed baseline and post-intervention assessments. Cognitive performance improved among all four groups, with no significant group × time effects. For academic achievement, there was no significant interaction effect between the combined intervention group and placebo. We encourage future studies in this neglected area in order to determine the most optimal design of school-based nutrition and PA programs to enhance overall cognitive performance.

## 1. Introduction

Executive functions (EFs) are integral to mental health and are essential for optimal social, psychological and academic development early in life [1]. Appropriate nutrition and engagement in regular physical activity (PA) have been shown to facilitate optimal cognitive development and function, including EFs [2]. Consequently, school health programs and physical education have been widely promoted with the intent of improving cognitive performance and academic achievement.

Growing evidence suggests that higher EFs are related to better academic achievement and increased chance to attend higher education [3], leading to greater economic success [4], and well-being later in life. At the same time, deficits in EFs early in life are attributable to a combination of poverty-related factors, such as low socioeconomic environment, poor health and limited access to basic resources.

EFs are understood as fundamental neurocognitive processes involved in information organization and processing, and language development. Accordingly, preschool years are a critical period of cognitive development with increases in EFs, such as information processing and inhibitory control [5]. Both information processing and inhibitory control contribute to more complex cognitive functions, such as problem solving, reasoning and planning, which set the trajectory for learning [1,3].

Children from low- and middle-income countries (LMICs) are especially at risk of food insecurity [6,7], multiple micronutrient deficiencies [8], malnutrition (including both overnutrition/obesity and undernutrition/stunting), as well as diseases such as soil-transmitted helminthic infections [9,10], which are linked to poor academic and cognitive performance [11]. Studies with schoolchildren from Ethiopia [12], Zimbabwe [13] and South Africa [14,15] show an association between monthly family income, academic achievement, and stunting. The South African cohort, in particular, showed that normal-weight girls achieved significantly better academic achievement than their stunted peers [14]. School feeding policies are an important contribution to daily dietary intakes, as micronutrient deficiencies are widespread in low-resourced areas. One in 2 beyond the National School Nutrition Programme [16] children under the age of 5 years suffers from some kind of micronutrient deficiency [17,18,19,20] with iron, iodine, vitamin A and zinc being the most common micronutrient deficiencies [4,19,21]. In South Africa, children from marginalized areas are prone to suffer from multiple deficiencies [8], in addition to stunting and obesity. With multi-micronutrient deficiencies as a risk factor for healthy brain development, nutrition programs, including micronutrient supplementation, may have the potential to enhance or compensate for deficits in cognitive performance. A recent review that investigated the effects of nutritional interventions on cognitive development among preschoolers found that children from LMICs who were nutritionally deprived showed significant improvements after iron and MMNS in cognitive performance [22], such as expressive language, problem-solving skills, working memory, and to a lesser degree, improvements in inhibitory control. In addition, the effect of MMNS on cognitive outcomes was positive for preschoolers from low quality schools. In contrast, no improvements were shown on growth parameters with MMNS independently of school quality [23]; moreover, particularly the youngest children seemed to benefit from the MMNS, while children above the age of 5 showed no such effects [24]; the authors, however, pointed out that caution should be exercised when interpreting the results due to the clinical heterogeneity of the studies. Another systematic review with schoolchildren in particular reported mixed results of fortified food on cognitive outcomes [25]; however, studies investigating the effects of micronutrient supplementation during primary school years (children aged 6–12 years) remain scarce.

Due to the concerning a rise in sedentary behavior together with non-communicable diseases, the World Health Organization (WHO) published a Global Action Plan in 2018, which recommended PA promotion already at an early age [26]; children also become considerably less physically active as they go through primary school [27], which creates an additional risk factor for cognitive development, making this a particularly relevant target population for health programs. Worldwide, a considerable shift in lifestyle factors is already responsible for the fact that 81% of 11–17-year-olds lack sufficient physical activity [26]. A previous meta-analysis confirmed the important role of PA on various cognitive domains in children, with the largest gains observed with longer interventions (22 weeks) and session duration of PA (30–60 min) [28]; however, these results were obtained from pediatric populations in high-income countries. PA interventions that were carried out in schools in marginalized neighborhoods showed promising, yet inconsistent, results [29,30]; for example, weekly PA lessons incorporated into the schools’ curriculum improved concentration capacity and academic achievement [29,30,31]. In Chile, a 30 min PA lesson before class (in addition to students’ weekly 2 h physical education classes) improved language and mathematics scores, but not cognitive performance [30].

Furthermore, the majority of the existing health interventions that aimed at improving cognitive performance have focused on a single type of intervention, such as PA or nutrition [32], and have primarily targeted children in their first 5 years of life; however, important processes of brain development continue during later childhood [33], and research is also needed to identify factors that impact on the cognitive performance of primary school children, particularly from marginalized areas. Additionally, it is unclear from the literature whether a combined MMNS and PA intervention would result in greater improvements in cognitive performance and, consequently academic achievement, than a single intervention.

To address this gap, the present study compared the effects of a 12-week school health program that included both a PA and MMNS intervention among 6-to-12-year-old children and was conducted in four South African primary schools located in marginalized, peri-urban communities. The body of research suggests that both interventions have independent effects that might have an additive value when combined. We hypothesized that implementing a combined school health program with a daily MMNS and 2 weekly PA sessions would positively influence children’s (i) cognitive performance (including information processing and inhibitory control), as well as (ii) their academic achievement (end of the year results); moreover, we were specifically interested whether (iii) a combined intervention would be superior to MMNS or PA alone.

## 2. Materials and Methods

### 2.1. Participants and Procedures

Participants were recruited from four quintile, three public primary schools (41 classes) located in a peri-urban area of Nelson Mandela Bay, Eastern Cape, South Africa. Schools in South Africa are classified into 5 quintiles, with quintile one being the poorest, and quintile 5 the least poor. Schools in quintile 1–3 are considered to be disadvantaged schools, and parents of children who attend these schools generally have poor ratings when it comes to their income and literacy levels. Prior to contacting schools, the appropriate school authorities were informed. The principals were provided with detailed information about the objectives, procedures, and risks/benefits of the study, after which they had a chance to express their interest. We informed parents/guardians prior to starting the baseline data collection that participation in this study is voluntary, that all data would be handled confidentially, and that participants could leave the study at any time without further obligation. Before data collection commenced, parents/guardians were asked to provide written informed consent and children were asked to provide oral assent.

We included schools if they had the facilities/space for physical education lessons, and did not participate in any other research or clinical trial. Children were included if (i) they attended grades one to 4 at baseline, (ii) were aged between 6–12 years, (iii) had written informed consent from a parent/guardian, (iv) did not participate in other study projects, nutritional programs beyond the National School Nutrition Programme [16], and clinical trials in the past 6 months and during the study period, and (v) had no medical conditions that would preclude participation in PA or the absorption of multi-micronutrients, as determined by qualified medical personnel. 

### 2.2. Study Design

The present study used data from a larger, double-blind, cluster-randomized controlled trial (KaziAfya). The KaziAfya study was designed to evaluate the effects of PA and MMNS on children’s growth, health and wellbeing in three African countries [34]. A sample size calculation based on a priori power analysis suggested the need to recruit around 1320 South African primary schoolchildren. In each of the four schools in the peri-urban area of Nelson Mandela Bay, classes from grades 1 to 4 were randomly assigned to the following intervention groups: (i) PA + MMNS, (ii) PA + placebo, (iii) MMNS, and (iv) the placebo group. The intervention was implemented between April 2019 and September 2019. As a result of the school holidays (mid-of-term break: June–July 2019), the actual intervention period is comprised of 12 weeks.

### 2.3. Randomization of Treatment Groups

The allocation of intervention groups occurred at the grade level to ensure that each intervention group was represented at all grade levels and in all schools (see Appendix A). The research officer who performed the randomization was not involved in the recruitment or data collection to guarantee double-blinding for data analyses. Children allocated to the PA group received 2 weekly 45-min PA lessons (one “moving to music” and one “physical education” lesson). The lessons were based on the KaziKidz toolkit, which was developed according to the South African curriculum in 2018 (see: https://www.kazibantu.org/kazikidz/ accessed on 24 April 2022). The physical education coach assisted the class teacher to implement PA lessons. Every morning, the children received an orange-flavored chewing tablet containing either multi-micronutrients or a placebo, which was distributed to them by the teacher. MMNS were formulated based on the MixMeTM powder sprinkle developed by DSM Nutritional Products South Africa (Johannesburg, South Africa) [35], which was modified to meet the current age-related intake recommendations (see Appendix A). Supplements were produced, packaged, tested for quality, and delivered to schools following the storage recommendations. Placebo tablets were administered to children in the PA and non-intervention conditions. Placebo tablets were produced using a similar process and contain cocoa powder, sugar, sorbitol, citric acid, and an orange flavor. Both active and placebo orange-flavored tablets were packaged in small containers of 60 tablets each. Independent of their group allocation, children benefited from health, hygiene and nutrition education lessons that were implemented as part of the Life Orientation Skills subject area.

### 2.4. Sample Size Calculation

The sample size was determined using a priori power analyses in G*power 3.1 (open source software; https://www.psychologie.hhu.de/arbeitsgruppen/allgemeine-psychologie-und-arbeitspsychologie/gpower accessed on 24 April 2022) [36,37], utilizing a 2 × 2 factorial design (MANOVA: Repeated measure, within-between interaction). The effect sizes on proposed outcomes from prior research were evaluated. MMNS interventions among preschoolers from marginalized communities yield small to medium–large effect sizes for inhibitory control *d* = 0.2 [23,38]; however, in the scarce literature on MMNS interventions for primary schoolchildren, reported effect sizes are smaller and largely inconsistent [22,38]. Research on (school-based) physical activity interventions on cognitive performance reveal similar large effect sizes, with an average reported effect size of *d* = 0.18 [28]. Therefore, a conservative effect size of *f* = 0.1 (*d* = 0.2) was used for the sample calculation. Based on this effect size, an alpha error probability of 0.05, and a correlation of 0.50 among repeated measures, the power analysis indicated 1095 participants to reach 80% statistical power. Considering a yearly dropout rate of 10% [39], the study recruited 1320 children per country (and *n* = 330 per intervention arm).

### 2.5. Adherence

The South African school system is compulsory from first grade on. Therefore, it was expected that absenteeism would not differ between the various groups.

### 2.6. Demographic and Anthropometric Data

Demographic data (age, sex) were collected at baseline and according to information on the teachers’ school register list. Nutritional status was determined based on the children’s weight and height and categorized by z-scores for stunting, underweight, overweight/obese and normal weight, using WHO child growth standards [40]. The weight of the schoolchildren was measured to the nearest 0.1 kg using a digital scale (Tanita; MC-580; Tanita Corp., Tokyo, Japan). Height was measured to the nearest 0.1 cm, using a stadiometer. Stunting was defined as height-for-age z-score (HAZ) <−2; underweight as weight-for-age z-score (WAZ) <−2; overweight as body mass index (BMI)-for-age z-score (BAZ) >1 in the absence of underweight; and normal weight as BAZ ≥−2 and BMI <1 in the absence of underweight.

### 2.7. Cognitive Performance

The computerized Flanker task was used to assess inhibitory control and children’s information processing as one domain of executive cognitive function. The task was programmed and administered with E-Prime 2.0 Software (Psychology Software Tools; Sharpsburg, PA, USA). Prior to starting the test, children received standardized verbal instructions from a trained researcher. The test was applied in the morning during a regular school day and at similar times at baseline and post-intervention.

Each child was seated in front of one laptop, with a total of 10 children in one room. Sets of stimuli were presented on a black screen and consisted of 5 white fishes (in a horizontal row, pointing to the right or left). Children were instructed to identify the direction of the central target fish of those sets. The central targeted fish was either pointing in the same direction (congruent trial; e.g., >>>>>) or opposite direction (incongruent trial; e.g., >><>>) than the flanking fishes. Children pressed the right arrow on the keyboard for a right direction and the left arrow for a left direction, using one finger. The total task took approximately 15 min to complete and contained 2 practice rounds with 60 trials in total and 2 rounds with 40 trials each, with a variation of the inter-trial interval between 1100 and 1500 ms to reduce the likeliness of guessing. A fixation of the stimuli lasted for 2500 ms. Congruent and incongruent trials were presented with equal probability and in a randomized order. The mean reaction time (correct responses only) and accuracy (proportion of correct responses) for each trial type (congruent; incongruent) were calculated. Performance on congruent and incongruent trials indicated information processing and inhibitory control, respectively. To ensure that only participants who understood the task correctly were included in analyses, datasets with accuracy rates lower than chance (≤50%) during the second practice round were removed.

### 2.8. Academic Achievement

The end-of-year results were provided by the school principals. With the baseline being conducted at the beginning of the academic year 2019, we used end-of-year results from the previous academic year 2018. Subsequently, with follow-up assessments taking place at the end of the academic year 2019, we used end-of-year results from the same year 2019 to describe academic achievement post-intervention. The mean of 2 core subjects (home language, mathematics) was used as an indicator of academic achievement. The South African school system uses a 7-point grading scale from 1 (1–29%, “not achieved”) to 7 (80–100%, “outstanding achievement”), with higher scores reflecting better academic achievement. 

### 2.9. Statistical Analyses

Descriptive statistics were calculated for all study variables (M (mean), SD (standard deviation), *n*, %). Only children with full baseline and post-intervention data in cognitive and academic outcomes were included. Normal distribution of sex, age, nutritional status, cognitive performance and academic achievement was explored with Kolmogorov–Smirnov tests separately for all four intervention groups. Given that these variables were not normally distributed among intervention groups (except for age), baseline differences between intervention groups were tested via non-parametric Kruskal–Wallis tests. In case of significant baseline differences, these variables were taken into account as covariates in the subsequent analyses. Separate linear mixed-model regression analyses were computed for each cognitive performance indicator and academic achievement to examine the effects of the intervention. These models included the outcome variable as a dependent variable, group as a fixed factor, and school classes as random effect. In the event of baseline imbalances in cognitive function and academic achievement, the analyses of intervention effects will be controlled for baseline levels. Pairwise comparisons will be conducted between the combined intervention and each single intervention and control. Mean and mean difference scores (with standard errors and 95% confidence intervals (CI) of the least significant differences are reported to examine the effects of the intervention. The level of statistical significance was set at *p* < 0.05 across all analyses. All analyses were performed with SPSS version 28 (IBM Corporation; Armonk, NY, USA) for Windows.

## 3. Results

### 3.1. Baseline Characteristics of Study Participants

A total of 1365 children between 6 and 12 years were enrolled in the KaziAfya trial in South Africa. Valid baseline and post-intervention data for cognitive performance and academic achievement, and valid age and sex data were available for 932 children. A flow diagram including numbers for valid data for each group can be found in Figure 1. Table 1 illustrates the baseline characteristics of the overall study population, as well as each intervention group. A significant difference in age was found between the groups, with the placebo group having the lowest mean age. Further, baseline differences between groups were observed for the main outcome variables (cognitive performance and academic achievement), with the lowest scores observed for the combined MMNS + PA group and the placebo group.

### 3.2. Intervention Effect on Cognitive Performance

Means and inferential statistics of the linear mixed models are presented in Table 2. No significant time × group differences were observed in reaction time and accuracy on either the congruent or incongruent trials after adjusting for age and school class. Over the course of the intervention, children from all intervention groups demonstrated significant improvements in reaction time and accuracy, both in inhibitory control (incongruent) and information processing (congruent) (Figure 2). Pairwise comparison did not reveal that the combined intervention (PA + MMNS) resulted in significantly higher cognitive performance than MMNS or PA alone in any of the cognitive performance indicators.

### 3.3. Intervention Effect on Academic Achievement

Means and inferential statistics of the linear mixed models are presented in Table 2. No significant time x group difference was observed for academic achievement after adjusting for age and school-class. Pairwise comparisons reveal that the combined intervention (PA + MMNS) resulted in significantly higher academic achievement than MMNS alone (*F*(1,16.78) = 22.45, *p* < 0.001), but not if compared to the PA (*F*(1,15.90) = 1.38, *p* = 0.257) or the placebo condition (*F*(1,18.74) = 0.108, *p* = 0.747). While both single-intervention groups showed a decline in academic achievement, end-of-year results were improved in the combined intervention group as well as the placebo group. There were no significant differences between MMNS and PA (*F*(1,14.85) = 2.96, *p* = 0.106).

### 3.4. Adverse Outcomes Associated with the Interventions

In the present study, only 18 students reported the following adverse effects, such as diarrhea, vomiting, and nausea, which were limited in duration and severity. At post-intervention, no additional side effects have been reported.

## 4. Discussion

The key finding was that a school-based physical activity and multi-micronutrient supplementation intervention did not benefit the cognitive performance and academic achievement of South African primary schoolchildren from disadvantaged schools compared to a placebo group. All groups improved cognitive performance, with no differences between treatment arms. Thus, indicating a similar learning effect in all four groups. Overall, age was the strongest determinant of performance outcome [41]. With regard to academic achievement, it appears that both the combined treatment group and the placebo group showed significant improvements.

This study was guided by 3 hypotheses, and each will now be addressed. First, it was hypothesized that MMNS would lead to improved cognitive performance outcomes and academic achievement compared to controls. In the present study, results were inconclusive, mirroring previous research conducted with primary school-aged children [23,24,29]. For example, a 5.7-months food supplementation intervention carried out with children above the age of 4 years did not have a significant effect on working memory; however, in the same study, a cohort of children aged 4 years and younger showed a beneficial effect [24]. Nevertheless, a recent study with South African primary schoolchildren aged 6–11 years has shown that micronutrient supplementation is also beneficial for cognitive performance among this age group; however, the intervention period was considerably longer with 8.5 months [42]. In the present study, the intervention was implemented over a period of 6 months, yet, as a result of school holidays, the actual intervention period was comprised of 12 weeks; however, a recent review of micronutrient supplementation interventions on cognitive performance has shown positive effects even for interventions lasting less than 3 months [22]. Previous research also revealed that long end-of-year school holidays lasting 2–3 months can reduce academic performance independently of any intervention [43]. In the present study, the school term-break lasted 3 weeks, during which the supplementation was paused. As a consequence, we cannot fully rule out that this interruption offset some of the potential benefits of our intervention on cognitive performance; however, a prior MMNS study revealed that a wash-out effect only occurs after 16 weeks [44]. We, therefore, do not consider the 3-week break to be a major flaw of the study design; moreover, it is important to note that any school-based health intervention is affected by holiday-related interruptions, since holiday-breaks are an integral part of the school calendar. For instance, physical education classes cannot be provided during school holidays. Likewise, it is not possible to continue daily micronutrient supplementation outside the schools, as we would not be certain (a) whether a child takes their supplements at home on a daily basis (b) or whether supplements would be shared between family members. Therefore, and despite potential issues associated with holiday-related interruptions, we deemed a school-based intervention still to be the best approach to reach a large number of primary school-aged children on a daily basis. We would also like to highlight that, because holiday-related interruptions were equal in all intervention arms and across all classes/schools, this confounding factor was controlled for in our trial; moreover, the term-break was followed by several further weeks of intervention, until the follow-up assessments were conducted. In addition, another study investigating the effects of an MMNS intervention among South African preschoolers observed improvements in cognitive functions after 11 weeks; these 11 weeks were interrupted by 1 week of school holidays, which indicates that a school-based intervention can be effective regardless of shorter interruption periods [45]. It should also be mentioned that within the present study population, additional analyses revealed an effect of MMNS on fat-free mass, indicating that the 3-week term-break did not modify the effect of supplementation at least on body composition [46].

Previous studies show that the effect of MMNS on cognitive performance was also more prominent in malnourished children [22]. Although children in the present study attended schools from marginalized communities, we do not have data regarding their level of micronutrient deficiency; it is also noteworthy that children from all groups performed well in the cognitive tests at baseline (accuracy congruent and incongruent outcomes). Therefore, it is possible that the lack of findings is due to a ceiling effect in the present sample; this assumption is supported, as the largest improvements in cognitive performance outcomes were observed among the 2 groups with the lowest baseline performance scores: the combined MMNS + PA group and the placebo group. In line with this observation, a micronutrient intervention trial with Indian preschoolers found improved language and inhibitory control, but only for children attending low-quality schools, which was associated with lower cognitive performance [23]; this highlights once more that the causes of poor growth and development are complex, and that other social and environmental factors may compromise the effectiveness of interventions in these settings [2,47,48]. Additionally, low socioeconomic status, lack of family support [2,49], infectious diseases [10], fetal alcohol syndrome [50,51], low nutrient availability [52] and being hungry or fatigued may hinder the age-related improvement in certain cognitive processes and academic achievement compared to well-nourished and healthy children [32,52]. Nevertheless, it remains a challenge for policymakers to determine the optimal nutritional intervention (where: setting, when: age group, and duration to achieve minimum effects) to promote optimal cognitive development in all school-aged children. 

Second, it was hypothesized that a PA intervention would improve cognitive performance outcomes and academic achievements compared to controls. Given that children from all groups improved their cognitive performance, we observed no additional beneficial effect of the PA intervention; however, substituting academic classes with 2 weekly PA sessions was neither detrimental to children’s cognitive performance nor academic achievement, but at the same time, reduced sedentary behavior and raised consciousness for an active and healthy lifestyle [53,54]. Nevertheless, the lack of statistically significant effects in the present study may indicate that the frequency and intensity of the PA intervention was not sufficient to have an impact; this conclusion was also made by authors of a similar school-based PA program among older South African primary schoolchildren that resulted in neither positive nor negative effects on selective attention and academic achievement compared to children attending regular academic classes [29]. A recent meta-analysis by Ludyga and colleagues [28] highlights that type, frequency, and intensity of exercise are important moderators and even affect different cognitive domains. For example, coordination exercises improve largely executive functions. While the present PA intervention included a variety of different exercise types (e.g., locomotor skills, coordination, sports, and games), frequency and intensity may have been inadequate; this conclusion is supported by looking at the frequency in effective PA programs: In a study with adolescents aged 12–15 years, beneficial effects on working memory and inhibitory control were reported following a daily 20-min exercise bout over 8 weeks [55]; however, environmental factors such as large class sizes can pose additional challenges with respect to the intensity and quality of physical education lessons [29]. The review by Pesce et al. [56] concluded that high-quality PA and the way such activities are carried out have a significant impact on cognitive outcomes. The lack of qualified physical education teachers at the primary school level in South African public schools is an important factor that needs to be addressed by the educational system.

Third, it was postulated that a combined MMNS and PA intervention would be superior to a single intervention. The data partly support this assumption in the sense that cognitive functions improved across all 4 intervention groups (Figure 2), whereas academic achievement slightly dropped among the 2 single interventions, but improved across the combined PA + MMNS and the placebo group. Yet, baseline academic achievement scores revealed significant group differences (*p* < 0.001), with the lowest end-of-year results in the combined PA + MMNS and placebo arm. In this respect, children with a lower initial exam grade might have had a greater chance to improve. The fact that baseline differences in academic achievement were found between treatment groups was an unexpected finding, although these schools were similar in size and student population. The present study showed that age was significantly different among all groups (*p* < 0.001), with children in the placebo group being the youngest. Among other unobserved factors, this might explain the significant baseline differences in academic achievement. In contrast, baseline accuracy scores for the cognitive performance were already quite high, with 92–94% for congruent trials (information processing) and 83–88% for incongruent trials (inhibitory control) (Table 2). To outweigh any test inclination strategy, it is necessary to evaluate improvements in accuracy scores in relation to reaction times (e.g., performance speed). At post-intervention, the largest improvements in reaction time were observed for the PA and placebo group, while the largest improvements in accuracy occurred in the combined intervention and the placebo group. Thus, the placebo group did not only increase performance speed, but also improved accuracy. The fact that the placebo group achieved similar results compared to the combined intervention group was unexpected and remains to be discussed. Nevertheless, a growing body of research shows that school health programs are particularly beneficial for at risk-populations, and that schools remain the ideal setting to reach these children via interventions [22,57,58].

This study has several strengths. First, the randomized factorial design enables us to consider each intervention, and the combined benefits of PA + MMNS intervention. Second, it contributes to covering the geographical gap in this area, being to the best of our knowledge, the first RCT to examine the effect of a combined PA and MMNS intervention on health outcomes, specifically on different domains of cognitive performance and academic achievement; moreover, most PA interventions examining effects on cognitive outcomes have been conducted in high-income countries, whereas the present study was carried out in marginalized public primary schools; this provides new insights into the feasibility of such school-based health programs in marginalized settings. Third, objectively validated and internationally accepted cognitive performance measures were applied in this study. Fourth, the supplementation was supervised by the class teacher to maximize adherence, and children of the placebo group were given a placebo tablet similar in taste and appearance to the supplement.

We are aware that this study is not without limitations. First, since PA lessons were held outdoors, it cannot be ruled out that the children of the placebo group also became more active during school hours; moreover, the exact time that a child spent in moderate and vigorous PA during the KaziKidz lessons is difficult to estimate, but would be important in order to provide in-school dose-response estimates for optimal cognitive development. Second, although we have controlled for baseline differences in cognitive functions (in line with previous research in this area [23,45]), we cannot be certain whether the micronutrient supplementation had an effect on children’s micronutrient status, which would require further blood analyses. As outlined in the study proposal [34], such analyses are planned, but these evaluations are not yet available due to delays caused by logistical difficulties in laboratory analysis of dried blood spots. Third, since the study involved children from peri-urban communities in South Africa, the results might not be applicable to urban or rural areas, children from different socioeconomic backgrounds, or even children from other LMICs. Fourth, although a clinical examination was conducted prior to the start of the intervention, children were not screened for any neurological disorders. Neurological disorders, such as attention deficit hyperactive disorder or fetal alcohol syndrome, are associated with impaired cognitive performance, and previous studies have reported a high prevalence amongst children from marginalized (peri-) urban communities in South Africa [50,51]. Fifth, given that a minority of children have access to a computer at home, this may be a disadvantage for some children during the computerized cognitive performance task (Flanker-task), despite the implementation of two practice rounds before the test.

## 5. Conclusions

The 12-week school health intervention showed no benefits of PA interventions and MMNS on cognitive and academic performance outcomes of primary school children when compared to the placebo group. Thus, the present study indicated that interventions, such as PA programs, which are known to benefit inhibitory control (in Western and high-income countries) cannot easily be transferred to primary school-aged children attending South African quintile 3 public schools. Future studies in this neglected area are encouraged so that the optimal design of nutrition programs for older children from marginalized areas can be determined, as well as the optimal design (type, frequency and intensity) of a school-based PA program to improve in-class attention and overall cognitive performance.

## Figures and Tables

**Figure 1 nutrients-14-02609-f001:**
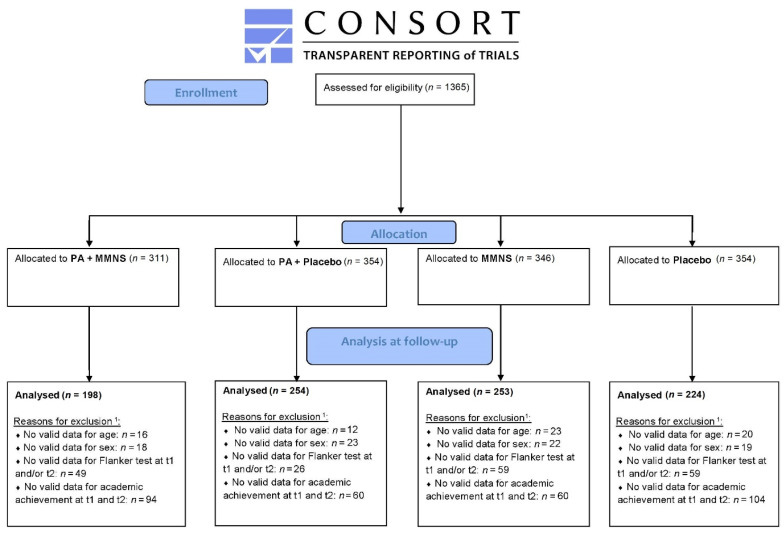
CONSORT flow diagram: progress of participants through the trial. Notes. ^1^ Children can be excluded for one or more reasons; e.g., a child with a missing Flanker test can also have a missing academic achievement. PA = Physical activity, MMNS = Multi-micronutrient supplementation, t1 = baseline, t2 = post-intervention.

**Figure 2 nutrients-14-02609-f002:**
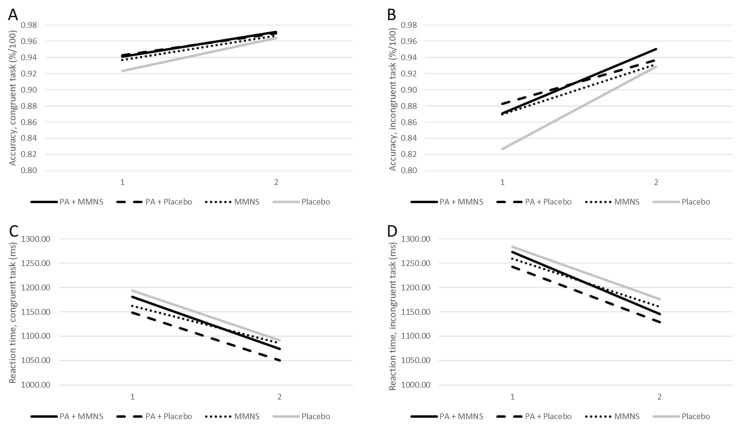
Accuracy and reaction time on the Flanker task displayed for baseline and post-intervention. Notes. Accuracy (congruent: (**A**); incongruent: (**B**)) and reaction time (congruent: (**C**); incongruent: (**D**)) on the Flanker task displayed for each intervention group for baseline (1) and post-intervention (2).

**Table 1 nutrients-14-02609-t001:** Descriptive characteristics and baseline differences between groups.

		Intervention Group				
Outcome Variables	Overall (*n* = 932)	PA + MMNS (*n* = 198)	PA + Placebo (*n* = 257)	MMNS (*n* = 253)	Placebo (*n* = 224)	*p*-Value
Participants characteristics						
Age, y, M (SD)	8.42 (1.94)	8.5 (1.22)	8.80 (2.95)	8.28 (1.47)	8.08 (1.29)	<0.001 *
Sex, girls, *n* (%)	458 (49.1)	105 (53.0)	136 (52.9)	119 (47.0)	98 (43.8)	0.126
Anthropometric measure						
Normal weight, *n* (%)	568 (73.6)	112 (67.5)	146 (73.0)	154 (73.3)	156 (79.6)	0.901
Stunting (HAZ < −2), *n* (%)	73 (9.5)	19 (11.4)	20 (10.0)	20 (9.5)	14 (7.1)	0.901
Overweight/obese (BAZ > 1), *n* (%)	131 (17.0)	35 (21.1)	34 (17.0)	36 (17.1)	26 (13.3)	0.901
Underweight (WAZ < −2), *n* (%)	62 (6.7)	13 (21.0)	21 (33.9)	15 (24.2)	13 (21.0)	1.000
Cognitive performance						
Accuracy, congruent, (%/100), M (SD)	0.94 (0.10)	0.94 (0.08)	0.94 (0.09)	0.94 (0.09)	0.92 (0.12)	0.206
Accuracy, incongruent, (%/100), M (SD)	0.86 (0.18)	0.87 (0.18)	0.88 (0.17)	0.87 (0.16)	0.83 (0.22)	0.073
Reaction time, congruent, ms, M (SD)	1170.16 (236.40)	1181.31 (217.88)	1148.82 (241.41)	1161.84 (229.74)	1194.19 (252.06)	0.301
Reaction time, incongruent, ms, M (SD)	1263.76 (264.64)	1273.67 (255.09)	1243.08 (257.03)	1259.07 (268.16)	1284.29 (277.16)	0.468
Academic achievement						
End of the year results, M (SD)	4.65 (1.23)	4.58 (1.29)	4.79 (1.31)	4.76 (1.24)	4.41 (1.02)	0.001 *
Language, M (SD)	4.58 (1.27)	4.54 (1.32)	4.72 (1.38)	4.70 (1.30)	4.31 (1.02)	0.001 *
Mathematic, M (SD)	4.71 (1.28)	4.62 (1.37)	4.86 (1.33)	4.82 (1.28)	4.51 (1.12)	0.005 *

Notes. PA = Physical activity, MMNS = Multi-micronutrient supplementation, M = mean, SD = standard deviation, y = year, HAZ = height-for-age z-score, BAZ = body mass index (BMI)-for-age z-score, WAZ = weight-for-age z-score. Kruskal Wallis test was used to compare all groups simultaneously. * *p* < 0.05.

**Table 2 nutrients-14-02609-t002:** Intervention effects on cognitive performance and academic achievement.

	Baseline	Post	Change Scores ^1^	Intervention Effect ^2^	
	M (SE)	M (SE)	M [95% CI]	Mean Difference (SE)	[95% CI]
Cognitive performance—information processing Accuracy, congruent (%/100)					
PA + MMNS	0.94 (0.01)	0.97 (0.01)	0.03 * [0.02; 0.04]	0.00 (0.008)	[−0.01; 0.02]
PA + Placebo	0.94 (0.01)	0.97 (0.01)	0.03 * [0.01; 0.04]	−0.00 (0.007)	[−0.02; 0.01]
MMNS	0.94 (0.01)	0.97 (0.01)	0.03 * [0.02; 0.04]	−0.00 (0.007)	[−0.02; 0.01]
Placebo	0.92 (0.01)	0.96 (0.01)	0.04 * [0.02; 0.06]	−	−
Reaction time, congruent (ms)					
PA + MMNS	1194.98 (33.10)	1087.63 (33.10)	−107.35 * [−144.01; −70.70]	−4.94 (24–37)	[−54.67; 44.80]
PA + Placebo	1161.63 (43.16)	1063.40 (43.16)	−98.23 * [−132.27; −64.18]	−3.13 (23.78)	[−51.85; 45.60]
MMNS	1164.86 (30.08)	1088.99 (30.08)	−75.87 * [−109.50; −42.23]	14.95 (23.63)	[−33.49; 63.38]
Placebo	1199.41 (26.22)	1096.41 (26.22)	−102.85 * [−137.37; −68.33]	−	−
Cognitive performance—inhibitory control					
Accuracy, incongruent (%/100)					
PA + MMNS	0.86 (0.02)	0.94 (0.02)	0.08 * [0.05; 0.11]	0.01 (0.016)	[−0.03; 0.04]
PA + Placebo	0.88 (0.02)	0.93 (0.02)	0.05 * [0.03; 0.08]	−0.01 (0.015)	[−0.05; 0.02]
MMNS	0.87 (0.02)	0.93 (0.02)	0.06 * [0.04; 0.09]	−0.01 (0.015)	[−0.04; 0.02]
Placebo	0.82 (0.02)	0.92 (0.02)	0.10 * [0.07; 0.13]	−	−
Reaction time, incongruent (ms)					
PA + MMNS	1278.72 (25.99)	1150.88 (25.94)	−127.84 * [−172.12; −83.56]	−22.51 (32.85)	[−89.51; 44.49]
PA + Placebo	1254.31 (41.84)	1140.44 (41.84)	−113.87 * [−153.20; −74.55]	−16.46 (32.19)	[−82.33; 49.42]
MMNS	1260.19 (27.95)	1161.55 (27.94)	−98.64 * [−139.20; −58.08]	−6.61 (32.00)	[−72.13; 58.91]
Placebo	1287.86 (26.14)	1179.67 (26.07)	−108.19 * [−149.04; −67.34]	−	−
Academic achievement End of year results					
PA + MMNS	4.39 (0.30)	4.45 (0.30)	0.06 [−0.14; 0.26]	−0.06 (0.20)	[−0.46; 0.34]
PA + Placebo	4.83 (0.13)	4.49 (0.13)	−0.34 * [−0.59; −0.09]	−0.31 (0.19)	[−0.71; 0.08]
MMNS	4.72 (0.18)	4.07 (0.18)	−0.65 * [−0.85; −0.45]	−0.72 (0.19) *	[−1.11; −0.33]
Placebo	4.39 (0.14)	4.61 (0.14)	0.23 * [0.03; 0.40]	−	−
Language					
PA + MMNS	4.33 (0.35)	4.45 (0.35)	0.33 [0.09; 0.56]	0.04 (0.18)	[−0.34; 0.41]
PA + Placebo	4.78 (0.15)	4.42 (0.15)	−0.39 * [−0.06; −0.15]	−0.32 (0.18)	[−0.69; 0.05]
MMNS	4.66 (0.21)	4.15 (0.21)	−0.63 * [−0.84; −0.43]	−0.55 (0.18) *	[−0.91; −0.18]
Placebo	4.28 (0.18)	4.47 (0.18)	0.23 * [0.02;0.44]	−	−
Mathematics					
PA + MMNS	4.46 (0.26)	4.46 (0.26)	0.01 [−0.23; 0.22]	−0.15 (0.28)	[−0.71; 0.42]
PA + Placebo	4.88 (0.12)	4.56 (0.12)	−0.32 * [−0.57; −0.06]	−0.28 (0.28)	[−0.84; 0.28]
MMNS	4.88 (0.16)	3.99 (0.16)	−0.79 * [−1.00; −0.57]	−0.87 (0.28) *	[−1.43; −0.31]
Placebo	4.51 (0.12)	4.75 (0.12)	0.24 * [0.03; 0.45]	−	−

Notes. PA = Physical activity, MMNS = Multi-micronutrient supplementation, SE = standard error, CI = confidence interval. ^1^ All estimates are from linear mixed models, including group (intervention groups; placebo) and timepoint (baseline; post-intervention) as fixed effects, school classes as random effect and age as covariate. Negative values indicate within-group improvement of cognitive performance. ^2^ All estimates of intervention effect (intervention condition—placebo condition) in the respective outcome variable (post-intervention) are from linear mixed models, including group as a fixed factor, school classes as random effect and age as covariate. Negative values for estimates indicate a group difference between a specific intervention condition compared to the placebo condition, with the intervention condition showing higher within-group improvement. * *p* < 0.05.

## Data Availability

The datasets are not publicly available due to the sensitivity of the data. Please contact the author if you have a request.

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
