# Peer review of "Evaluation of a Physical Activity and Multi-Micronutrient Intervention on Cognitive and Academic Performance in South African Primary Schoolchildren"

_nutrients, 2022, doi:10.3390/nu14132609_

Round 1

Reviewer 1 Report

  1. The introduction can be improve with a literature review on the nutrition and nutritional status of children.
  2. According to the reviewer, too few inclusion and exclusion criteria were used. It also wonders if the age range of children is not too wide in the context of the assessed indicators

  3. I believe that the impact of supplementation cannot be assessed without quantifying the supply of micronutrients in the diet. In my opinion, this is a serious methodological error that makes it impossible to draw conclusions.

  4.  

    Moreover, a serious drawback of the work is the lack of assessment of the nutritional status of children mainly blood tests and the level of selected minerals and vitamins). In the opinion of the reviewer, the assessment of the nutritional status only on the basis of the BMI level is absolutely insufficient.

  5.  

    The methodology did not indicate the initial level of physical activity. Did the children differ in motor skills, levels of physical activity or not? It seems to me that the initial differentiation in this regard could be decisive for the final results of the experiment.

  6. With regard to the above comments, the conclusions formulated by the authors may not be truthful and may result in inappropriate recommendations regarding the role of physical activity and diet supplementation for parents and guardians of children.

Author Response

Thank you for your interest and constructive feedback. We have addressed all further points of criticism, and hope that we were able to modify the manuscript to your satisfaction.

Reviewer 2 Report

Overall, it is a good intervention study and well conducted and well written.

Introduction

1.      Well written and it provides addition to current knowledge.

Methods:

2.      This study has many limitations. Three months of study period is not sufficient to improve cognitive performance of primary school children, hence it is necessary to explain why it was conducted for 3 months; may be logistics reasons.

3.      Usually cross over trail is the best study design for this type of study. However, it is not clear whether this is a secondary data analysis or selecting a sample from the previous study and conducted the new trail. Please specify it.  

4.      Computerized flanker test is well performed by the children who engaged with computer games. Is there any information on that? If so please include, otherwise just mention it in the discussion considering the usual pattern among children in these schools. 

5.      Reference of the original study is not indicated in the text, may be printing error.

Results

6.      Table 1: Better to include percentage of underweight (thinness) also.

7.      There may be difference of cognitive performance between children who are thin and not. If it is significant, try to add a table for that.

8.      Academic achievement and accuracy is lowest in placebo group. However, after 12 weeks placebo group has significant difference.  Better to add reasons for baseline difference.

9.      Placebo group is nutritionally also better, add the explanation. 

Discussion

10.   Better to include above facts quoting suitable references.  

Conclusion

11.   It is adequately addressed the study.

Author Response

(The authors gave the same response as above.)

Round 2

Reviewer 1 Report

Thank you for the explanations and correct correction. Unfortunately, I cannot agree that due to randomization it is assumed that there are no differences between the conditions regarding intake of micronutrients in the diet. I do not change my position on the issue of methodological shortcomings. I am aware that the methodological correction at this stage is not possible, so I leave the decision on accepting the manuscript to the editor.

Author Response

We appreciate your opinion and all previous made comments to improve the manuscript. 
